# Long-Term Effects of Rifaximin on Patients with Hepatic Encephalopathy: Its Possible Effects on the Improvement in the Blood Ammonia Concentration Levels, Hepatic Spare Ability and Refractory Ascites

**DOI:** 10.3390/medicina58091276

**Published:** 2022-09-14

**Authors:** Keiji Yokoyama, Hiromi Fukuda, Ryo Yamauchi, Masashi Higashi, Takashi Miyayama, Tomotaka Higashi, Yotaro Uchida, Kumiko Shibata, Naoaki Tsuchiya, Atsushi Fukunaga, Kaoru Umeda, Kazuhide Takata, Takashi Tanaka, Satoshi Shakado, Shotaro Sakisaka, Fumihito Hirai

**Affiliations:** 1Department of Gastroenterology and Medicine, Faculty of Medicine, Fukuoka University, 7-45-1 Nanakuma, Jonan-ku, Fukuoka-shi 814-0180, Fukuoka, Japan; 2Higashi Hospital, 593-1 Hirotsu, Yositomi-machi, Chikujo-gun 871-0811, Fukuoka, Japan

**Keywords:** rifaximin, hepatic encephalopathy, hyperammonemia, refractory ascites, blood renin levels

## Abstract

*Background and Objectives*: To investigate the long-term efficacy of rifaximin (RFX) for hyperammonemia and efficacy for refractory ascites in patients with cirrhosis. *Materials and Methods*: We enrolled 112 patients with liver cirrhosis who were orally administered RFX in this study. Changes in the clinical data of patients were evaluated up to 36 months after RFX administration. The primary endpoint was a change in blood ammonia levels. Secondary endpoints included changes in clinical symptoms, Child–Pugh (CP) score, number of hospitalizations, degree of refractory ascites, adverse events, and the relationship between RFX administration and the renin-angiotensin-aldosterone system. *Results*: An improved rate of overt hepatic encephalopathy (HE) of 82.7% was observed 3 months after RFX administration, which significantly induced a progressive decrease in blood ammonia concentration and an improved CP score up to 36 months. No serious RFX treatment-related adverse events were observed. 36.5% in patients after RFX administration improved refractory ascites. After RFX administration, patients with satisfactory control of hepatic ascites without addition of diuretic had lower renin concentration than those with poor control (*p* < 0.01). At less than 41 pg/mL renin concentration, the control of refractory ascites was significantly satisfactory (*p* < 0.0001). *Conclusions*: RFX reduced blood ammonia concentration and improved hepatic spare ability and the quality of life of patients with long-term HE to up to 36 months. Our study revealed the effects of RFX against refractory ascites, suggesting that renin concentration may be a predictive marker for assessing ascites control.

## 1. Introduction

Hepatic encephalopathy (HE) is a serious complication of severe hepatic dysfunction, often caused by acute liver failure, liver cirrhosis (LC), or a neuropsychiatric symptom caused by portosystemic shunt formation; 30–45% of patients with LC develop overt HE (OHE) [1]. The development of HE can be attributed to low fatty acid concentration and the systemic presence of ammonia, mercaptans, and amines (trimethylamines), which are mainly nitrogen-containing compounds obtained from dietary proteins or produced by gut microbiota during gastrointestinal bleeding. Ammonia plays a central role in the activity of these encephalopathy-inducing compounds [2,3]. In addition, changes in the intestinal flora composition and an increase in the proportion of ammonia-producing bacteria during LC and portal hypertension have been shown before [4,5].

In clinical practice, the West-Haven criteria, which consist of grades from minimal (or I) to IV, are used internationally to classify the degree of impaired consciousness [6,7]. According to the International Hepatic Encephalopathy and Nitrogen Metabolism Society criteria, HE is classified as OHE when clear neuropsychiatric symptoms (grades II-IV) are observed and as covert HE (CHE) when clear neuropsychiatric symptoms (minimal or grade I) are absent.

Rifaximin (RFX), a rifamycin-based antibacterial agent obtained by the culture of the actinomycete *Streptomyces mediterranei*, has a broad antibacterial spectrum covering gram-positive, gram-negative, aerobic, and anaerobic bacteria. RFX is an antibiotic that elicits its effect by inhibiting bacterial RNA synthesis. When administered orally, its intestinal absorption is extremely low (<0.4%), restricting antibacterial activity mainly in the intestinal tract, with few systemic side effects [8,9,10].

RFX has been recommended by the American Association for the Study of Liver Diseases and the European Association for the Study of the Liver guidelines for the treatment of HE [6,7]. The efficacy of RFX following 12 weeks of administration has been confirmed through 37 institutional prospective randomized trials (Phase II/III and Phase III trials) in Japan [11]. Recently, evidence-based clinical practice guidelines for LC in Japan reported that “Since non-absorbable antimicrobial agents are effective of HE both in initial or recurrent episodes, its administration is a basic treatment for HE (Recommendation: strong, 100% agreed, evidence level A)” [12]. Therefore, the assessment of more cases and evaluation of the efficacy of long-term RFX administration in clinical settings may be valuable. Therefore, in this study, we aimed to evaluate the efficacy of RFX in the treatment for hyperammonemia and the secondary effects linked to its long-term administration.

## 2. Materials and Methods

The inclusion criterion was adult patients with LC aged 20 years or older with hyperammonemia who started oral RFX treatment (1200 mg/day) between January 2017 and December 2021. A total of 142 patients were initially enrolled in this study, of which 30 patients were excluded according to the following criteria: patients who were not tracked for more than a month (*n* = 12), patients without LC (*n* = 5), Patients who had already administered kanamycin (*n* = 5), patients with apparent poor oral compliance (*n* = 4), patients who discontinued oral administration for reasons other than side effects within a month (*n* = 3), and patients who dropped out of the study (*n* = 1). Finally, 112 patients were included in the final analysis (Figure 1).

We evaluated changes in the clinical data of each patient up to 36 months after RFX treatment was initiated. Observations were terminated when RFX administration was discontinued or when the patient died. The median observation period was 838 days and the primary endpoint was a change in blood ammonia concentration. Secondary endpoints included changes in clinical symptoms, Child–Pugh (CP) score, number of hospitalizations, improvement of refractory ascites, adverse events, and the relationship between RFX administration and the renin-angiotensin-aldosterone system (RAAS). Refractory ascites was defined as moderate or higher ascites retention or resistance to treatment with loop diuretics and/or anti-aldosterone diuretics. The improvement of refractory ascites (Satisfactory control) was defined as a decrease of at least one grade in the ascites CP score.

### 2.1. Statistical Analyses

All analyses were performed using JMP version 13.0 (SAS Institute, Charlotte, NC, USA). *p* < 0.05 was considered statistically significant. Results are presented as the mean ± standard deviation. Changes in clinical data were calculated using the Wilcoxon signed-rank test. Differences between the two independent groups were analyzed using Fisher’s exact test and the Mann–Whitney U test.

The relationship between RFX administration and the RAAS was determined using a receiver operating characteristic (ROC) curve. The optimal cut-off value for an independent variable that most accurately predicted the dependent variable was identified using the area under the curve (AUC) and Youden’s index. Multivariate analysis was performed using the Cox proportional hazards model to estimate the hazard ratios (HRs) for events. Multiple logistic regression was performed using factors with *p* < 0.05 in the univariate analysis to identify factors contributing to the effects of satisfactory control of refractory ascites without diuretic use 3 months after RFX treatment.

### 2.2. Ethical Statement

The study protocol was approved by the ethics committee of Fukuoka University Hospital (approval number: H20-07-005). The study was conducted in compliance with the principles of the Declaration of Helsinki of 1975 (revised in 2013) and the Ethical Guidelines for Medical Research of the Ministry of Health, Labor and Welfare. The data collected were anonymized. As this was a retrospective study conducted using past medical information, it was not possible to obtain consent from the enrolled patients. Therefore, a waiver for informed consent and permission to opt out of the study was obtained from the ethics committee. This study was approved by the ethics committee of our university, as detailed on our website (http://www.med.fukuoka-u.ac.jp/research/life_med_ethic/ accessed on 10 July 2020).

## 3. Results

### 3.1. Clinical Characteristics of the Study Population

The clinical characteristics of the study population are summarized in Table 1. A total of 112 patients (75 men and 37 women) with LC were included, with an average age of 65.1 years. The CP class A, B, and C subjects were 6, 56, and 50, respectively. Model for end-stage liver disease (MELD) score were 12.9 ± 4.43 and MELD sodium score were 13.4 ± 4.98, respectively. The West-Haven grade minimal or I, II, III, and IV patients were 50, 38, 22, and 2, respectively. A total of 40 patients had comorbid hepatocellular carcinoma (HCC). A total of 42 and 59 patients had treatment histories of HCC and esophageal or gastric varices, respectively. Only two patients had previously been treated for HCC and were clearly tumor-free at the time of study enrollment, one patient after hepatectomy and the other patient after percutaneous radiofrequency ablation (RFA). A total of 43, 56 and 11 patients had already been administered loop diuretics and anti-aldosterone diuretics and non-selective beta blockers (NSBBs), respectively. RFX was added with continuation of medications already prescribed to treat hyperammonemia (Table 2).

### 3.2. Long-Term Effects of Administration of RFX on Hyperammonemia and HE

Mean blood ammonia concentration before and after 1, 3, 6, 12, 18, 24, 30, and 36 months of RFX treatment were 116.9, 77.2, 80.1, 77.9, 80.1, 74.4, 79.7, 83.0, and 84.5 μg/dL, respectively (Figure 2). The difference in blood ammonia concentration before and after RFX administration was significant at all time points. The rate of the improvement of OHE after administering RFX for 1 week, 1 month, and 3 months was 58.3%, 78.7%, and 82.7%, respectively.

### 3.3. Secondary Therapeutic Effects

Mean CP scores before and after 1, 3, 6, 12, 18, 24, 30, and 36 months of RFX treatment were 9.54, 8.60, 8.33, 8.26, 8.14, 8.0, 8.34, 7.59, and 7.81, respectively (Figure 3). Mean serum albumin concentration before and after 1, 3, 6, 12, 18, 24, 30, and 36 months of RFX treatment were 2.87, 2.92, 2.99, 3.00, 3.10, 3.12, 2.99, 3.26, and 3.34 g/dL, respectively (Figure 4). Mean prothrombin activity before and after 1, 3, 6, 12, 18, 24, 30, and 36 months of RFX treatment were 59.4%, 61.5%, 62.7%, 61.8%, 63.1%, 65.5%, 61.7%, 70.6%, and 67.9%, respectively (Figure 5). Mean total-bilirubin before and after 1, 3, 6, 12, 18, 24, 30, and 36 months of RFX treatment were 3.30, 2.59, 2.06, 2.15, 1.93, 1.56, 1.73, 1.18, and 1.50 mg/dL, respectively (Figure 6). The difference in CP scores before and after RFX administration was significant at all time points, while the differences in serum albumin concentration, prothrombin activity, and total-bilirubin concentration were partially significant at each points.

In the 3 months before and after RFX administration (*n* = 97), the number of hospitalizations due to liver-related events included for ascites and HE significantly decreased from 0.8 to 0.2 times (*p* < 0.001) (Figure 7). Seventy-four patients had refractory ascites before starting RFX. After RFX administration for 3 months, refractory ascites improved in 27 of the 74 patients (36.5%). The clinical characteristics of these 27 patients are shown in Appendix A. In many cases, the treatment of refractory ascites was performed in parallel with other therapies. Therefore, we also assessed if other therapies such as diuretics, BCAA, carnitine, zinc, abstinence, steroid, and partial splenic embolization (PSE) used with RFX administration had any effects on refractory ascites. It is noteworthy that six patients showed improvements in refractory ascites without the addition of other treatments, especially diuretics. Furthermore, patients with CP class A and B showed significantly better control of refractory ascites 3 months after oral RFX without addition of diuretic than those with CP class C (*p* < 0.001) (Appendix A).

### 3.4. Adverse Events

No patients showed pseudomembranous colitis. Loose or watery stools were observed in 3.6% of patients (4/112 patients), abdominal discomfort was observed in 1.8% of patients (2/112 patients) and 0.9% of patients (1/112 patients) experienced nausea immediately after RFX administration. However, all the adverse events improved without medication. No serious RFX treatment-related adverse events were observed.

### 3.5. Relationship between RFX Administration and the RAAS

The course of control of ascites was analyzed in 81 patients whose serum renin and aldosterone concentration were examined before RFX treatment. Three months later, patients with satisfactory control of ascites without addition of diuretic had significantly lower serum renin concentration than those with poor control (*p* = 0.001) (Figure 8).

Based on this result, univariate and multivariate analyses of factors affecting satisfactory control of refractory ascites without addition of diuretic 3 months after RFX administration were performed. The results are presented in Table 3. In univariate analysis, significant differences were found in low CP scores (HR, 20.4; 95% confidence interval (CI), 2.29–243; *p* < 0.01) and low serum renin concentration (HR, 26.6; 95% confidence interval (CI), 3.46–551; *p* < 0.01). In the multivariate analysis, not only these two factors, but also Etiology (Not alcohol), MELD score, MELD sodium score, T-bil, and EGV treatment history (Absence), which are likely confounding factors at *p* < 0.2, were included. Multivariate analyses revealed that low CP scores (HR, 62.7; 95% confidence interval (CI), 1.5–4526; *p* = 0.028) and low serum renin concentration (HR, 48.4; 95% CI, 4.6–1483; *p* < 0.01) were associated with satisfactory control of refractory ascites. In contrast, high CP scores, and high serum renin concentration, were important indicators of poor control of ascites after RFX therapy. The ROC curve for patients with satisfactory control of refractory ascites and low renin concentration revealed the following: AUC = 0.779; cut-off value: 41 pg/mL; sensitivity: 71.4%; specificity: 79.5%; positive predictive value (PPV): 78.9%; negative predictive value (NPV): 72.1% (Figure 9). At 41 pg/mL or higher renin concentration, the control of refractory ascites was significantly poor (*p* < 0.0001) (Table 4).

## 4. Discussion

In this study, we found that long-term administration of RFX for 36 months progressively reduced blood ammonia concentration, suppressed OHE, and improved the CP score in Japanese patients. In addition, no serious side effects, such as pseudomembranous colitis, were observed, and oral RFX was found to be safe.

A previous study reported the safety and efficacy of the long-term administration of RFX for 24 months or longer in the treatment of HE [13]. Other observational studies and meta-analyses have reported that the addition of RFX to the treatment regimen comprising synthesized disaccharides significantly reduced the risk of the recurrence of OHE compared to synthesized disaccharide monotherapy without compromising tolerability [14,15]. In another study with a small sample size, long-term administration of RFX for 5 years was shown to be associated with a reduced risk of developing complications, such as portal hypertension, and improved survival in 23 patients with alcohol-related decompensated cirrhosis [16].

Previous small-scale studies have shown RFX treatment to improve complications due to LC in Japanese patients, including an increased survival rate 6 to 12 months after RFX administration, reduced blood ammonia concentration, and an improved HE and nutritional status [17,18,19]. Our large-scale study corroborates previous studies and establishes that long-term administration of RFX is safe in the Japanese population. Secondary effects after RFX treatment are improved or maintained hepatic reserve, significantly reduced numbers of hospitalizations, and hospital inpatient days; therefore, RFX treatment improves the quality of life of patients with HE. Some studies have reported the effects of RFX on CP score retention [20] and medical cost reduction [21,22,23]. In addition, the role of RFX in improving CHE has been reported [24,25,26]; RFX is expected to be used in the future for the comprehensive management of patients with LC and the control of blood ammonia concentration and HE.

After 3 months of RFX administration, refractory ascites improved in 36.5% of the patients. Furthermore, refractory ascites improved in six patients without the addition of diuretic. In most patients with refractory ascites, diuretics were added at the same time as starting RFX treatment, making clear evaluation difficult; however, improvement in these six patients confirms the role of RFX in improving refractory ascites. A previous study reported that treatment with RFX in 50 patients with cirrhosis and refractory ascites significantly reduced fasting body weight and ascites [27]. In another study, the effects of RFX and midodrine added to diuretic therapy were evaluated. These combination therapies enhanced diuresis in patients with refractory ascites and improved systemic and renal hemodynamics [28]. RFX has also been shown to improve systemic hemodynamics in decompensated cirrhosis [29]. Additionally, portal pressure and endotoxin activity were reduced following the administration of RFX in combination with NSBBs [30]; improvements in portal hypertension may contribute to the reduction of refractory ascites. Furthermore, RFX has been reported to suppress the onset of acute kidney injury and hepatorenal syndrome [31]. In conclusion, the findings of this study and previous studies establish that RFX administration can reduce refractory ascites.

However, there exist reports of significant and non-significant effects of RFX on the improvement of refractory ascites, therefore, it is necessary to examine the predictors of these effects further. The findings of our study suggested that 41 pg/mL or lower renin concentration before RFX administration could be predictive of the ascites-improving effect of RFX.

Aldosterone concentration does not seem to be a determinant of RFX effects, as no significant difference was found in aldosterone concentration before and after RFX treatment, probably because anti-aldosterone diuretics were administered in many patients (50.0%). In patients with LC, portal hypertension and endogenous endotoxin concentration increase due to bacterial-translocation-induced higher vasodilation, as well as underfilling and constriction of renal blood vessels, all of which are known to be activated by the RAAS [32,33,34]. Renin is a proteolytic enzyme synthesized mainly in juxtaglomerular cells and active renin acts on angiotensinogen to produce angiotensin I. The RAAS plays an important role in the regulation of blood pressure and water-electrolyte metabolism in the body. Previous studies have reported that angiotensin II promotes the proliferation of activated stellate cells in a dose-dependent manner and that AT-1-receptor expression is markedly increased in patients with LC [35,36]. Moreover, in patients with LC, high renin concentration has been reported to be associated with a poor prognosis and renin has been suggested to be an indicator of portal hypertension and increased risk of ascites [37]. Previous studies have also reported that RAAS inhibitors suppress the progression of liver fibrosis [38,39,40] and that direct renin inhibitors and selective aldosterone inhibitors suppress liver fibrosis progression in nonalcoholic steatohepatitis (NASH) [41,42].

Although high renin concentration reflects high endogenous endotoxin concentration, there have been no previous reports detailing the relationship between serum renin concentration and HE. Consequently, when comparing serum renin concentration and the presence of HE before RFX administration in this study, there was a tendency for patients with OHE to have high serum renin concentration before RFX treatment, but the difference was not statistically significant (*p* = 0.22). Future studies with a larger sample will be warranted. In a previous study, RFX was shown to significantly improve endotoxin activity and HE without significantly affecting the diversity of gut microbiota [43]. Therefore, the decrease in endotoxin activity due to RFX administration is considered to be one of the main factors in improving refractory ascites, and the relationship between RFX treatment and RAAS must be further studied.

In this study, serum renin concentration in 38 patients before and 3 months after RFX initiation was compared. We found that RFX did not significantly decrease serum renin concentration (*p* = 0.393). However, RFX administration has been reported to reduce cardiac output and increase systemic vascular resistance, with a decrease in plasma renin activity [44]. Therefore, renin concentration in association with RFX treatment should be studied in a larger sample to determine the mechanism of action of RFX.

This study had several limitations. First, its retrospective, almost single-center study design made it difficult to compare our findings with those of prospective studies. Second, there was no control group in the present study. Almost all patients with hyperammonemia were started on RFX treatment, therefore, there were very few patients who would be eligible as the control group with hyperammonemia. Third, several studies have reported changes in gut microbiota after RFX administration [45,46]; however, we did not analyze gut microbiota which may be associated with changes in endogenous endotoxin concentration. Nonetheless, considering that only a few studies have been carried out involving more than 100 patients, our findings provide a valuable evaluation of the efficacy, long-term administration effects, and secondary effects of RFX in the treatment of hyperammonemia.

## 5. Conclusions

We demonstrated that RFX improved blood ammonia concentration and hepatic spare ability of patients with HE in the long-term up to 36 months. Moreover, our results suggested the favorable effects of RFX against refractory ascites. Serum renin concentration levels may be useful as a marker for assessing ascites control and may be further investigated.

## Figures and Tables

**Figure 1 medicina-58-01276-f001:**
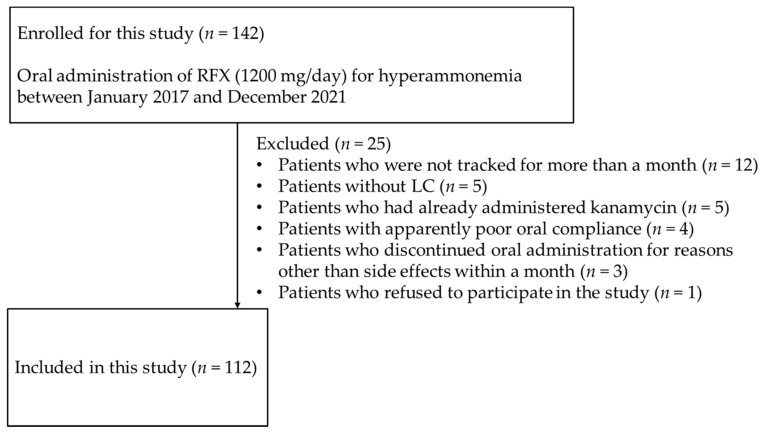
Flowchart illustrating patient selection for the study. *n*, number; RFX, rifaximin.

**Figure 2 medicina-58-01276-f002:**
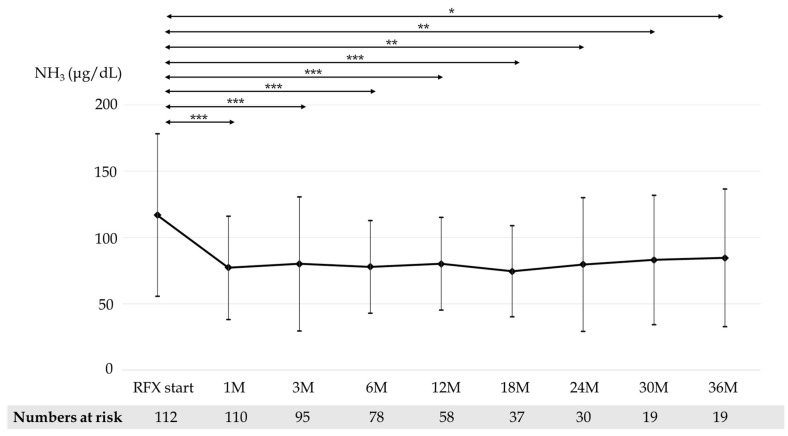
Changes in mean blood ammonia concentration before and 1, 3, 6, 12, 18, 24, 30, and 36 months after RFX administration. *p*-values were determined using the Wilcoxon signed-rank test (*p* < 0.05 *, *p* < 0.01 **, *p* < 0.001 ***). RFX, rifaximin.

**Figure 3 medicina-58-01276-f003:**
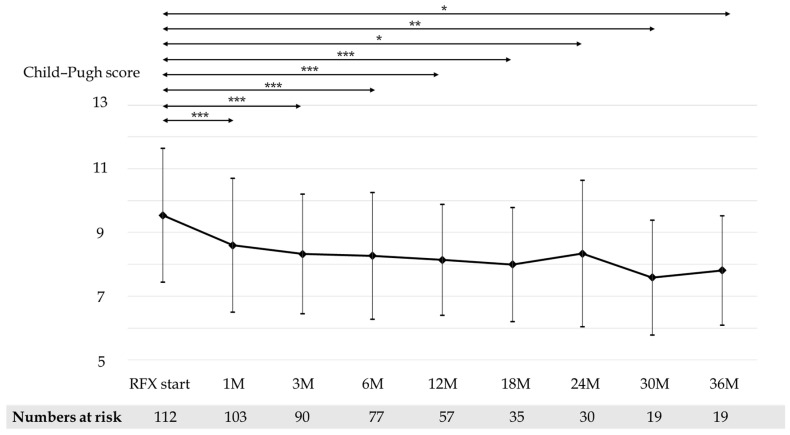
Changes in mean Child–Pugh score before and 1, 3, 6, 12, 18, 24, 30, and 36 months after RFX administration. *p*-values were determined using the Wilcoxon signed-rank test (*p* < 0.05 *, *p* < 0.01 **, *p* < 0.001 ***). RFX, rifaximin.

**Figure 4 medicina-58-01276-f004:**
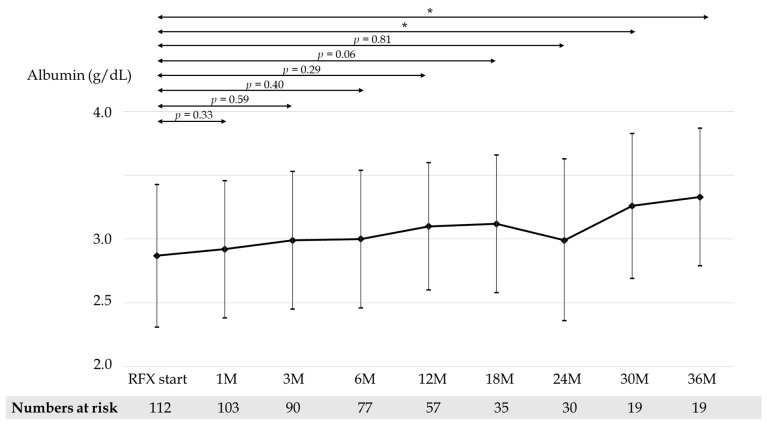
Changes in mean serum albumin concentration before and 1, 3, 6, 12, 18, 24, 30, and 36 months after RFX administration. *p*-values were determined using the Wilcoxon signed-rank test (*p* < 0.1 tendency differences, *p* < 0.05 *). RFX, rifaximin.

**Figure 5 medicina-58-01276-f005:**
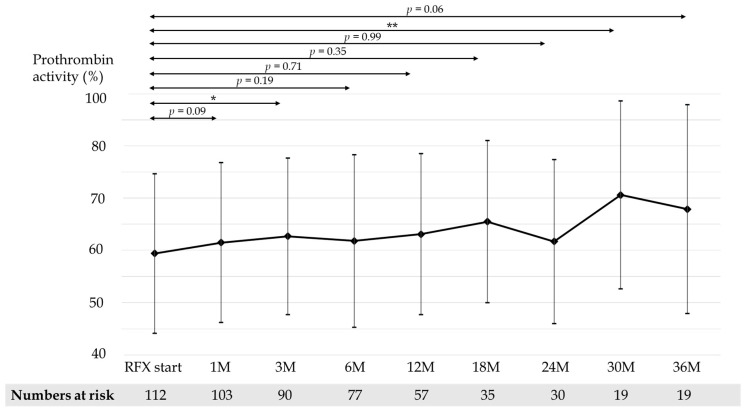
Changes in mean prothrombin activity before and 1, 3, 6, 12, 18, 24, 30, and 36 months after RFX administration. *p*-values were determined using the Wilcoxon signed-rank test (*p* < 0.1 tendency differences, *p* < 0.05 *, *p* < 0.01 **). RFX, rifaximin.

**Figure 6 medicina-58-01276-f006:**
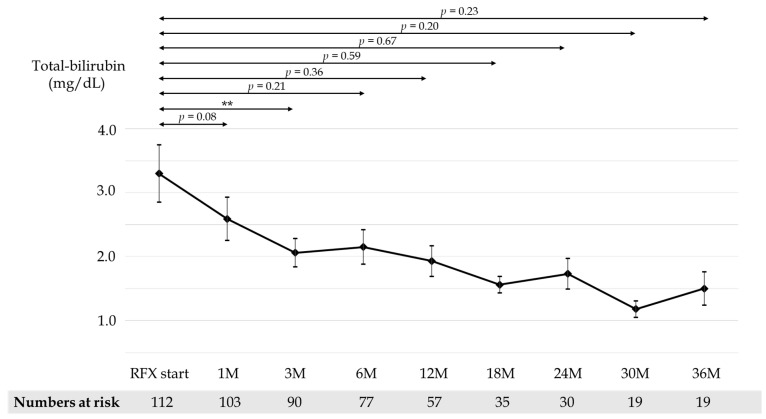
Changes in mean total-bilirubin before and 1, 3, 6, 12, 18, 24, 30, and 36 months after RFX administration. *p*-values were determined using the Wilcoxon signed-rank test (*p* < 0.1 tendency differences, *p* < 0.01 **). RFX, rifaximin.

**Figure 7 medicina-58-01276-f007:**
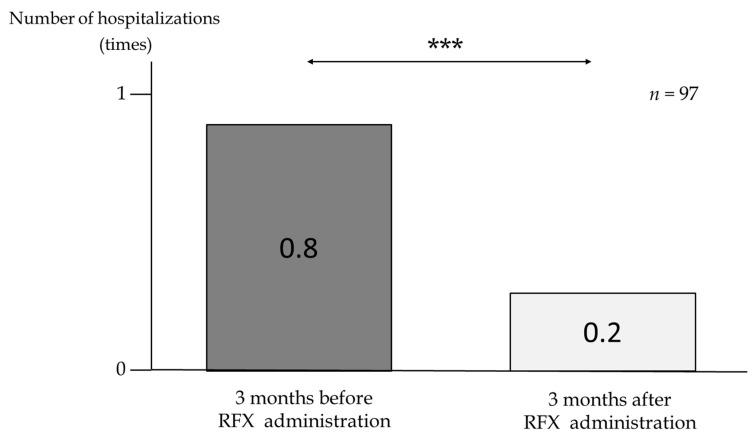
Number of hospitalizations due to liver-related events included for hepatic ascites and hepatic encephalopathy (*n* = 97) 3 months before and after RFX administration. *p*-values were determined using the Mann–Whitney U test (*p* < 0.001 ***). *n*, number; RFX, rifaximin.

**Figure 8 medicina-58-01276-f008:**
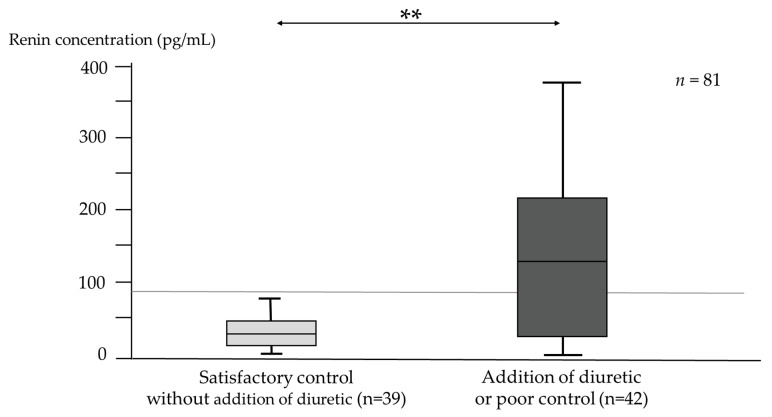
Relationship between control of refractory ascites 3 months after RFX administration and serum renin concentration before RFX administration (*n* = 81). *p*-values were determined using the Mann–Whitney’s U test (*p* < 0.01 **). *n*, number; RFX, rifaximin.

**Figure 9 medicina-58-01276-f009:**
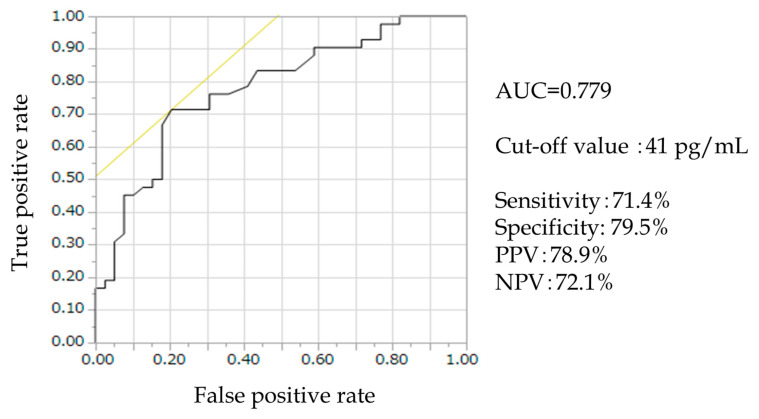
ROC curve of patients with satisfactory control of refractory ascites and corresponding renin concentration. AUC = 0.779; Cut-off value, 41 pg/mL; sensitivity, 71.4%; specificity, 79.5%; PPV, 78.9%; NPV, 72.1%. AUC, area under the curve; NPV, negative predictive value; PPV, positive predictive value; ROC, receiver operating characteristic.

**Table 1 medicina-58-01276-t001:** Clinical characteristics of the study population.

Number of Patients	112
Sex (male/female)	75/37
Age (in years, expressed as mean ± SD)	65.1 ± 11.6
Etiology	
Alcohol consumption	47 (42.0%)
Viral hepatitis	27 (24.1%)
NASH	14 (12.5%)
Alcohol consumption and viral hepatitis	7 (6.3%)
Alcohol consumption and autoimmune hepatitis	1 (0.9%)
Others	16 (14.3%)
Child–Pugh classification	
Class A	6 (5.4%)
Class B	56 (50.0%)
Class C	50 (44.6%)
Child–Pugh score (mean ± SD)	9.54 ± 2.1
MELD score	12.9 ± 4.43
MELD sodium score	13.4 ± 4.98
Blood ammonia concentration, NH_3_ (μg/dL) (mean ± SD)	117.0 ± 61.3
Prothrombin time, PT (%) (mean ± SD)	59.4 ± 15.3
Prothrombin time-international normalized ratio, PT-INR (mean ± SD)	1.36 ± 0.23
Serum albumin concentration, Alb (g/dL) (mean ± SD)	2.87 ± 0.57
Total-bilirubin, T-bil (mg/dL) (mean ± SD)	3.30 ± 0.45
Blood urea nitrogen, BUN (mg/dL) (mean ± SD)	16.3 ± 6.82
Serum creatinine concentration, Cr (mg/dL) (mean ± SD)	0.94 ± 0.69
Estimated glomerular filtration rate, eGFR (mL/min/1.73 m^2^) (mean ± SD)	69.6 ± 23.4
Serum sodium concentration, Na (mEq/L) (mean ± SD)	139.2 ± 3.04
West-Haven grade	
Minimal or I	50 (44.6%)
II	38 (33.9%)
III	22 (19.6%)
IV	2 (1.8%)
HCC	
Presence	40 (35.7%)
Absence	72 (64.3%)
History of HCC treatment	
Presence	42 (37.5%)
Absence	70 (62.5%)
History of gastrointestinal variceal treatment	
Presence	59 (52.7%)
Absence	53 (47.3%)
Administration of diuretics	
Loop diuretics	43 (38.4%)
Anti-aldosterone diuretics	56 (50.0%)
Administration of NSBBs	11 (9.8%)

HCC, hepatocellular carcinoma; MELD, model for end-stage liver disease; NASH, nonalcoholic steatohepatitis; NSBBs, nonselective beta blockers; SD, standard deviation.

**Table 2 medicina-58-01276-t002:** Patient pretreatment drug history for hyperammonemia treatment.

Pretreatment Drug	Number of Patients
Oral BCAA preparations	78
Synthetic disaccharides	47
Intestinal regulators and laxatives	38
Carnitine preparations	30
Zinc preparations	10

BCAA, branched chain amino acid; RFX, rifaximin.

**Table 3 medicina-58-01276-t003:** Univariate and multivariate analyses of factors involved in satisfactory hepatic ascites control 3 months after RFX administration (*n* = 81).

	Satisfactory Control *n* = 39	Poor Control *n* = 42	Univariate		Multivariate	
			HR 95% CI	*p*-Value	HR 95% CI	*p*-Value
Sex						
Male	27	30				
Female	12	12		0.83		
Age	66.8 ± 1.9	64.1 ± 1.8		0.32		
Etiology						
Alcohol	17	27				
Not alcohol	22	15		0.06		0.40
CP score	8.74 ± 0.30	9.90 ± 0.29	20.4 2.29–243	<0.01 **	62.7 1.5–4526	0.028 *
MELD score	11.7 ± 0.69	13.3 ± 0.68		0.10		0.92
MELD sodium score	12.1 ± 0.76	13.9 ± 0.74		0.10		0.83
NH_3_	123 ± 9.9	124 ± 9.5		0.96		
Alb	3.00 ± 0.08	2.85 ± 0.08		0.21		
T-bil	2.07 ± 0.67	3.59 ± 0.65		0.11		0.98
BUN	15.8 ± 1.14	16.6 ± 1.11		0.65		
Cr	0.91 ± 0.10	0.90 ± 0.10		0.93		
eGFR	72.8 ± 3.48	66.9 ± 3.39		0.22		
OHE						
Presence	24	21				
Absence	15	21		0.29		
HCC						
Presence	16	12				
Absence	23	30		0.24		
History of EGV treatment						
Presence	18	26				
Absence	21	16		0.16		0.13
Administration of anti-aldosterone diuretics						
Presence	15	22				
Absence	24	20		0.21		
Renin	40.9 ± 18.5	127.9 ± 17.8	26.6 3.46–551	<0.01 **	48.4 4.6–1483	<0.01 **
Aldosterone	264.4 ± 89.8	307.2 ± 80.4		0.72		

*p*-values were determined using Fisher’s exact test and the Cox proportional hazards model (*p* < 0.05 *, *p* < 0.01 **). Alb, Albumin; BUN, blood urea nitrogen; CI, confidence interval; CP, Child–Pugh; Cr, Creatinine; eGFR, estimated glomerular filtration rate; EGV, esophagogastric varices; HR, hazard ratio; HCC, hepatocellular carcinoma; NH_3_, Ammonia; OHE, overt hepatic encephalopathy; T-bil, Total-bilirubin.

**Table 4 medicina-58-01276-t004:** Relationship between hepatic ascites control 3 months after RFX administration and renin concentration (*n* = 81).

Renin Levels	Satisfactory Control	Poor Control
Less than 41 pg/mL	32	14
41 pg/mL or higher	7	28

*n*, number.

## Data Availability

Not applicable.

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
