# Peer review of "Long-Term Effects of Rifaximin on Patients with Hepatic Encephalopathy: Its Possible Effects on the Improvement in the Blood Ammonia Concentration Levels, Hepatic Spare Ability and Refractory Ascites"

_medicina, 2022, doi:10.3390/medicina58091276_

Round 1
Reviewer 1 Report
This manuscript reported the efficacy of RFX for long term administration. And authors suggesting that renin levels may be a predictive marker for assessing ascites control after RFX administration. The conclusion is very interesting for the readers. However, reviewer have some comments for this manuscript.
Major
1. The authors discussed that anti-aldosterone diuretics affects serum aldosterone concentration; however, it also affects serum renin concentration. Therefore, reviewer thought that Table 5 should include the presence or absence of anti-aldosterone diuretics in the multivariate regression analysis.
2. The serum renin concentration is an interesting indicator of ascites control. Reviewer suggest to evaluate the effect of serum renin concentration on hepatic encephalopathy. And if possible, please evaluate the evaluate the effect of serum renin concentration on prognosis.
Minor
1. In recent years, the term "Child grade A is no used, but Child Pugh A, or Child class A is used. Please correct the text and Table.
2. Please check the explanation in Table 4 and the text. Authors mentioned that there are a total of 28 cases without diuretics, however, in Table 4 showed that there are 40 cases. Reviewer think that "without diuretics" is unnecessary.
3. Please check the explanation in Table 8 and the text as well. Reviewer think that "without diuretics" is unnecessary.
4. Authors discussed that the association between RFX and endotoxin, and the association between RFX and renin. If there are any literature, please discuss the association between endotoxin and renin in the discussion session.
Author Response
We are grateful to Referee 1 for critical comments and useful suggestions that have helped us to improve our paper considerably. As indicated in the responses that follow, we have taken all these comments and suggestions into account in the revised version of our manuscript.
Comments by Referee 1.
First, considering suggestion of Referee 2, we excluded 5 cases of received kanamycin before RFX and reanalyzed the data as a total of 112 patients.
Major
1-The authors discussed that anti-aldosterone diuretics affects serum aldosterone concentration; however, it also affects serum renin concentration. Therefore, reviewer thought that Table 5 should include the presence or absence of anti-aldosterone diuretics in the multivariate regression analysis.
Response.
We reanalyzed our multivariate regression analysis in Table 5 to include the presence or absence of anti-aldosterone diuretics.
2-The serum renin concentration is an interesting indicator of ascites control. Reviewer suggest to evaluate the effect of serum renin concentration on hepatic encephalopathy. And if possible, please evaluate the evaluate the effect of serum renin concentration on prognosis.
Response.
In this study, there was a tendency for patients with overt encephalopathy to have high serum renin concentration before RFX treatment, but the difference was not statistically significant (P = 0.22). Therefore, we have added the following sentence to the Discussion section; ' Although high renin concentrations reflect high endogenous endotoxin levels, there have been no previous reports detailing the relationship between serum renin concentrations and hepatic encephalopathy. Consequently, comparison with serum renin concentration and the presence of HE before RFX administration in this study, there was a tendency for patients with overt encephalopathy to have high serum renin concentration before RFX treatment, but the difference was not statistically significant (P = 0.22). Future studies with a larger sample will be warranted.’.
Unfortunately, we have not conducted a detailed examination of the prognosis in this study, and we would appreciate the possibility of making this an issue for our next study.
Minor
1-In recent years, the term "Child grade A is no used, but Child Pugh A, or Child class A is used. Please correct the text and Table.
Response.
We have changed 'grade' to 'class' in the text and Table.
2- Please check the explanation in Table 4 and the text. Authors mentioned that there are a total of 28 cases without diuretics, however, in Table 4 showed that there are 40 cases. Reviewer think that "without diuretics" is unnecessary.
Response.
We have moved table 4 into suppl Material. We have matched the number of cases in the text and in the tables, and also eliminated "without diuretics".
3-Please check the explanation in Table 8 and the text as well. Reviewer think that "without diuretics" is unnecessary.
Response.
We have checked and eliminated "without diuretics".
4- Authors discussed that the association between RFX and endotoxin, and the association between RFX and renin. If there are any literature, please discuss the association between endotoxin and renin in the discussion session.
Response.
We have discussed the association between endotoxin and renin in the following sentences in the discussion section; 'In patients with LC, portal hypertension and endogenous endotoxin level increase due to bacterial translocation induced higher vasodilation, as well as underfilling and constriction of renal blood vessels, all of which are known to be activated by the RAAS [32–34]'. We hope you will confirm it.
Reviewer 2 Report
Yokoyama et al. describe a relatively large cohort of patients with LC receiving RFX as a longterm treatment for up to 36 months. They report on the effects of RFX treatment on ascites development and recurrence as well as HE development and respective hospital admissions with a special regard to renin levels. Despite such a large cohort being potentially interesting to study over such a long period of RFX treatment, the present study and form of presentation has several major weaknesses.
First of all, the study design is not adequate. I do understand that data was obtained retrospectively. With regard to the fact that RFX is not recommended in all LC patients, a control group of patients not receiving RFX should be available. The entire presented data cannot adequately be interpreted if not compared to a matching cohort of patients who did not receive RFX.
Additionally, the aims of the study are not sufficiently described. A more detailed and structured explanation is required.
The data presentation lacks several very important facts (see in detail below). Why is MELD not reported? Why is INR not reported? What was the rationale to investigate renin if the time course of retention parameters is not even presented? Use of NSBB is not reported eventhough this is crucial when reporting on renin-levels. Presented data on etiology is insufficient. It remains furthermore unclear why the patients received RFX? Did they all have HE at the time of study enrollment or just a history of HE? Details on additional treatments for HE are lacking. Information on history of TIPS, LTX or variceal bleeding is missing. It is also inadequate to only focus on hospital admissions due to HE or ascites and not paying attention to other forms of hepatic decompensation. Data on HCC is not adequately presented. The statistical procedure is not sufficiently explained and contains errors. Also, the visuell data presentation is not adequate. Please pay close attention to the detailed remarks below.
Abstract:
The part 'results' is incomprehensible and cannot be interpreted in this form. Comparisons and basic backgroundinformation is lacking.
Material and Methods:
- including only patients above 20 appears random and without justification. Please specify.
- there is no control group. This is major concern and makes all results more or less uninterpretable. This is a major mistake in study design
- what was the rationale to put these patients on RFX? Hyperammonemia is no indication to put patients on RFX.
Statistical analysis
- 'Results are presented as the mean ± standard deviation' - mean is not always the best option. Respective data distribution has to be considered. Please adjust
- 'The optimal cut-off value for an independent variable that most accurately predicted the dependent variable was identified using the area under the curve (AUC)' - This is to vague. What method has been used?
Results
- the patients who received kanamycin before RFX have to excluded since the prior antibiotic treatment certainly influences any observed effect of RFX
- table 1: please indicate the respective percentages, otherwise this table is hard to interpret and analyze
- table 1: I have never seen that only HCC treatment is reported and no information on HCC itself! I urgently advise to indicate if HCC is present at the time of enrollment and if history of HCC is present. History of HCC treatment can additionally be reported, if so, please indicate since treatment for HCC varies from surgery, transplantation, CTX, Y-Kinase inhibitors to BCS, it cannot be summarized into one group
- table 1: 'Etiology (there is some overlapping)' - this is an absolutely inadequate presentation and unprofessional. If there is overlapping, indicate and specify.
- table 2: it remains unclear if the patients remained on additional drugs that can reduce ammonia levels (dissacharides, BCAAs, LOLA). Please specify, especially in the results part.
- part '3.3. Secondary therapeutic effects' has to be changes. Please refer to the respective figures for exact numbers. This paragraph is otherwise unreasonable.
- figures 2-6: the presentation via box-plot is not ideal. I recommend changing to curves, still indicating respective SD's
- Why is the prothrombin activity given and not INR?
- Why are there no p-values provided for bilirubin after 3 months?
- Figure 7: I do not see a reason to limit the presented data to hospitalizations due to ascites and HE. What about ACLF? Variceal bleeding? Any other form of decompensation? All liver related events should be included in this analysis!
- table 3: put into suppl. material.
- table 3: concerning the HCV patients: where these patients with active HCV or cirrhotic after DAA or other treatment? Please indicate?
- table 4: the definition of poor and satisfactory control is insufficient.
- table 4: i do not see a reason to present the data in this form. Only CP A+B and C are compared. Of course, CP C patients have a poorer control. Where is the link to RFX? If it all, put table 4 into suppl. material
- figure 8: please indicate the connection to RFX treatment. It is otherwise not at all surprising or new that renin-levels are different between these groups.
- table 5: please assess MELD score and MELD sodium score as well
- table 5: please indicate all p-values. It is not always reasonable to focus only on the parameters which are significant in the univariate analysis since possible confounders are ignored that way. Please discuss and rework.
- table 5: what kind of variable selection was used (forward, reverse etc.?)
- please specify the calculation of the cut-off value and how this was defined.
Author Response
We are grateful to Referee 2 for critical comments and useful suggestions that have helped us to improve our paper considerably. As indicated in the responses that follow, we have taken all these comments and suggestions into account in the revised version of our manuscript.
Comments by Referee 2.
First of all, the study design is not adequate. I do understand that data was obtained retrospectively. With regard to the fact that RFX is not recommended in all LC patients, a control group of patients not receiving RFX should be available. The entire presented data cannot adequately be interpreted if not compared to a matching cohort of patients who did not receive RFX.
Response.
This is exactly what you have indicated. However, I am sure you also already understand that this is a retrospective study. Since most patients with hyperammonemia are administrated RFX in our department, there are very few patients who would be eligible as the control group with hyperammonemia. Therefore, we have decided to clearly specify it in the LIMITATION in this study. We added the following sentence in the Discussion section (limitation part); “Second, there was no control group in the present study. Almost patients with hyperammonemia were started RFX treatment, therefore, there were very few patients who would be eligible as the control group with hyperammonemia.”
Additionally, the aims of the study are not sufficiently described. A more detailed and structured explanation is required.
Response.
Because this study included a case in which RFX was administered to patients with hyperammonemia without overt hepatic encephalopathy, we have changed the statement of purpose in the text "effect on hepatic encephalopathy" to "effect on hyperammonemia.
The data presentation lacks several very important facts (see in detail below). Why is MELD not reported? Why is INR not reported? What was the rationale to investigate renin if the time course of retention parameters is not even presented? Use of NSBB is not reported even though this is crucial when reporting on renin-levels. Presented data on etiology is insufficient. It remains furthermore unclear why the patients received RFX? Did they all have HE at the time of study enrollment or just a history of HE? Details on additional treatments for HE are lacking. Information on history of TIPS, LTX or variceal bleeding is missing. It is also inadequate to only focus on hospital admissions due to HE or ascites and not paying attention to other forms of hepatic decompensation. Data on HCC is not adequately presented. The statistical procedure is not sufficiently explained and contains errors. Also, the visuell data presentation is not adequate. Please pay close attention to the detailed remarks below.
Response.
We have described our response to each of the points indicated below. Thank you in advance.
Abstract:
-The part 'results' is incomprehensible and cannot be interpreted in this form. Comparisons and basic background information is lacking.
Response.
We modified the text in the part 'results' section in Abstract as follows; 'An improved rate of overt hepatic encephalopathy (HE) of 82.7% was observed 3 months after RFX administration, which significantly induced a progressive decrease in blood ammonia levels and an improved CP score up to 36 months. No serious RFX treatment-related adverse events were observed. 36.5% in patients after RFX administration improved refractory ascites. After RFX administration, patients with satisfactory control of hepatic ascites without addition of diuretic had lower renin levels than those with poor control (P<0.01). At less than 41 pg/mL renin levels, the control of refractory ascites was significantly satisfactory (P<0.0001).'.
Material and Methods:
- including only patients above 20 appears random and without justification. Please specify.
Response.
This study included only adults 20 years or older, and we added "adult patients" in the text.
- there is no control group. This is major concern and makes all results more or less uninterpretable. This is a major mistake in study design
Response.
This is exactly what you have indicated. However, I am sure you also already understand that this is a retrospective study. Since most patients with hyperammonemia are administrated RFX, there are very few patients who would be eligible as the control group with hyperammonemia. Therefore, we have decided to clearly specify it in the LIMITATION in this study. We added the following sentence in the Discussion section (limitation part); “Second, there was no control group in the present study. Almost patients with hyperammonemia were started RFX treatment, therefore, there were very few patients who would be eligible as the control group with hyperammonemia.”
- what was the rationale to put these patients on RFX? Hyperammonemia is no indication to put patients on RFX.
Response.
This study also included patients with covert hepatic encephalopathy with hyperammonemia (West-Haven grade Minimal or I, n=51).
Statistical analysis
- 'Results are presented as the mean ± standard deviation' - mean is not always the best option. Respective data distribution has to be considered. Please adjust.
Response.
Because of the relatively large overall sample size and few extreme outliers in this study, we determined that mean values were more appropriate than median or most frequent values.
- 'The optimal cut-off value for an independent variable that most accurately predicted the dependent variable was identified using the area under the curve (AUC)' - This is to vague. What method has been used?
Response.
We used Youden’s index. We modified the text as follows; 'The optimal cut-off value for an independent variable that most accurately predicted the dependent variable was identified using the area under the curve (AUC) and Youden's index'.
Results
- the patients who received kanamycin before RFX have to excluded since the prior antibiotic treatment certainly influences any observed effect of RFX.
Response.
Considering your suggestion, we excluded 5 cases of received kanamycin before RFX and reanalyzed the data as a total of 112 patients.
- table 1: please indicate the respective percentages, otherwise this table is hard to interpret and analyze.
Response.
We added the respective percentages in the table 1.
- table 1: I have never seen that only HCC treatment is reported and no information on HCC itself! I urgently advise to indicate if HCC is present at the time of enrollment and if history of HCC is present. History of HCC treatment can additionally be reported, if so, please indicate since treatment for HCC varies from surgery, transplantation, CTX, Y-Kinase inhibitors to BCS, it cannot be summarized into one group.
Response.
In the current study, we had described cases of HCC comorbidity as almost synonymous with "History of HCC treatment," which we felt should be changed. We examined the current HCC comorbidities in more detail and found 40 cases, which we added to Table 1. Only two patients had previously been treated for HCC and were clearly tumor free at the time of study enrollment, one patient after hepatectomy and the other patient after RFA. Details of the two cases were added in the text.
- table 1: 'Etiology (there is some overlapping)' - this is an absolutely inadequate presentation and unprofessional. If there is overlapping, indicate and specify.
Response.
As you indicated, we thought it should be changed. We have included the details of the duplicate cases in the table 1.
- table 2: it remains unclear if the patients remained on additional drugs that can reduce ammonia levels (dissacharides, BCAAs, LOLA). Please specify, especially in the results part.
Response.
We modified the text as follows; 'RFX was added with continuation of medications already prescribed to treat hyperammonemia (Table 2).'.
- part '3.3. Secondary therapeutic effects' has to be changes. Please refer to the respective figures for exact numbers. This paragraph is otherwise unreasonable.
Response.
We have adjusted and modified each of the numbers. In the present study, among Figures 2-6, only Figure 2 has a slightly higher number of cases in which only the change of ammonia value was measured, so the number is slightly increased.
- figures 2-6: the presentation via box-plot is not ideal. I recommend changing to curves, still indicating respective SD's.
Response.
We changed to curves in figures 2-6.
- Why is the prothrombin activity given and not INR?
Response.
In the present study, we mainly included the change in prothrombin activity, which is commonly used in Japan, instead of INR as a parameter of the Child-Pugh score.
- Why are there no p-values provided for bilirubin after 3 months?
Response.
In the submitted manuscript, we omitted the description of P values without significant differences, but in this revision, we have newly added them in the figure.
- Figure 7: I do not see a reason to limit the presented data to hospitalizations due to ascites and HE. What about ACLF? Variceal bleeding? Any other form of decompensation? All liver related events should be included in this analysis!
Response.
For clarity in relation to the present study, 'hospitalizations for liver-related events' and 'hospitalizations for ascites and HE ' were described as almost synonymous. Therefore, we have changed it to 'hospitalizations due to liver-related events included for ascites and HE'.
- table 3: put into suppl. material.
Response.
We have moved Table 3 to the Supplementary Material.
- table 3: concerning the HCV patients: where these patients with active HCV or cirrhotic after DAA or other treatment? Please indicate?
Response.
All six HCV patients included in Table 3 were cirrhotic at the time of study enrollment. Four of the six patients were introduced to DAA in cirrhosis and achieved SVR; two patients had not yet been introduced to DAA. Unfortunately, the percentage of impact on hepatic reserve in the two patients with alcohol comorbidity was not known. At least none of the patients ongoing DAA at the time the data were obtained.
- table 4: the definition of poor and satisfactory control is insufficient.
Response.
In the Materials and Methods section, we described as follows; 'Refractory ascites was defined as moderate or greater ascites retention or resistance to treatment with loop diuretics and/or anti-aldosterone diuretics. Improvement of refractory ascites (Satisfactory control) was defined as a decrease in ascites CP score by at least one level.'.
- table 4: I do not see a reason to present the data in this form. Only CP A+B and C are compared. Of course, CP C patients have a poorer control. Where is the link to RFX? If it all, put table 4 into suppl. Material.
Response.
In the present study, the effect of RFX administration itself on ascites control was linked to CP score, without adding or increasing diuretics. Before evaluating the link with renin levels, which we want to emphasize most in the later section, we first examined the results with CP score, which is the most likely indicator. In fact, we believe that the results were as expected. The reason we chose CP A+B is that the number of CP A cases was extremely small (n=6), making it difficult to evaluate CP A alone. We have moved Table 4 to the Suppl. Material.
- figure 8: please indicate the connection to RFX treatment. It is otherwise not at all surprising or new that renin-levels are different between these groups.
Response.
In the present study, we emphasize that there are cases in which RFX administration itself, without additional or increased diuretics, has a beneficial effect on hepatic ascites control and is linked to renin levels before RFX administration. Although it might be expected, we believe that this study proved the point.
- table 5: please assess MELD score and MELD sodium score as well.
Response.
We additionally evaluated the MELD score and the MELD sodium score.
- table 5: please indicate all p-values. It is not always reasonable to focus only on the parameters which are significant in the univariate analysis since possible confounders are ignored that way. Please discuss and rework.
Response.
We have included not only renin levels and CP score, which were significantly different in univariate analysis, but also Etiology (Not alcohol), MELD score, MELD sodium score, T-bil, History of EGV treatment (Absence), which tended to be confounders at P<0.2, were added to the multivariate analysis. And we added the text as follows in the result section; 'In univariate analysis, significant differences were found in low CP scores (HR, 20.4; 95% confidence interval (CI), 2.29–243; P<0.01) and low blood renin levels (HR, 26.6; 95% confidence interval (CI), 3.46–551; P<0.01). In the multivariate analysis, not only these two factors, but also Etiology (Not alcohol), MELD score, MELD sodium score, T-bil, and EGV treatment history (Absence), which are likely confounding factors at P<0.2, were included.'
- table 5: what kind of variable selection was used (forward, reverse etc.?)
Response.
We made the variable selection used forward. Therefore, the odds ratios are reversed compared to the first submission. Namely, this odds ratio is that the lower the renin concentration and CP score, the better the control of ascites.
- please specify the calculation of the cut-off value and how this was defined.
Response.
As for the cutoff values for the ROC curve, we used Youden’s index as described above.